# Does the Application of Additional Hydrophobic Resin to Universal Adhesives Increase Bonding Longevity of Eroded Dentin?

**DOI:** 10.3390/polym14132701

**Published:** 2022-06-30

**Authors:** Graça Maria Abreu Pereira de Brito, Daniella Oliveira Silva, Rayssa Ferreira Cavaleiro Macedo, Michel Wendlinger Cantanhede Ferreira, Jose Bauer, Flavia de Brito Pedroso, Alessandra Reis, Fabiana Suelen Figuerêdo Siqueira, Alessandro Dourado Loguercio, Andres Felipe Millan Cardenas

**Affiliations:** 1Department of Postgraduate Program in Dentistry, CEUMA University, São Luis 65075-120, Brazil; graca_ma@yahoo.com.br (G.M.A.P.d.B.); daniella.ortho@gmail.com (D.O.S.); fabiana.siqueira@ceuma.br (F.S.F.S.); andres.cardenas@ceuma.br (A.F.M.C.); 2Department of Postgraduate Program in Dentistry, Federal University of Maranhao, São Luis 65085-805, Brazil; rayssafcm@gmail.com (R.F.C.M.); bauer@ufma.br (J.B.); 3Department of Pharmacology, State University of Ponta Grossa, Uvaranas 84030-900, Brazil; michelwendlinger@gmail.com (M.W.C.F.); flasbrito@hotmail.com (F.d.B.P.); 4Department of Restorative Dentistry, State University of Ponta Grossa, Uvaranas 84030-900, Brazil; alereis@uepg.br

**Keywords:** bond strength, eroded dentin, hydrophobic, adhesive–dentin interface, nanoleakage

## Abstract

This paper evaluates the effect of an additional hydrophobic resin coat (extra HL) associated with universal adhesives on sound and eroded dentin and evaluated immediately or after 2 years of water storage to improve the microtensile bond strength (μTBS) and nanoleakage (NL) when compared to the use of universal adhesives only. Sixty-four molars were assigned to eight groups using the following combinations: 1. dentin substrate, including sound and eroded dentin; 2. treatment, including the control and extra HL and storage time (immediately and after two-years of storage). Two universal adhesives (Prime & Bond Active or Scotchbond Universal) were evaluated. Before restoration, half of the teeth were subjected to soft-drink erosion. Composite buildups were bonded; specimens were stored (37 °C/24 h), sectioned into resin–dentin bonded sticks and tested for microtensile bond strength and nanoleakage using SEM (immediately and after two-years of storage). Three-way ANOVA and Tukey’s test (α = 0.05%) were used. In the immediate testing, the application of extra HL did not increase microtensile bond strength values compared with the control group in either substrate (*p* > 0.05). However, extra HL significantly decreased nanoleakage values when applied to eroded and sound dentin (*p* = 0.0001). After two years, the application of extra HL produced significantly higher microtensile bond strength and lower nanoleakage values than the control group for both adhesives (*p* = 0.0001). In all cases, sound dentin showed higher microtensile bond strength and lower nanoleakage values than eroded dentin (*p* = 0.000001). An extra HL increased the bond strength and reduced nanoleakage in eroded dentin after two-years of storage.

## 1. Introduction

The bonding performance and predictability of adhesives are challenging for eroded dentin surfaces [1,2,3,4]. Continuous acid action can be induced by biological and chemical alterations, jeopardizing the restorative properties of dental materials [2,3].

Physical barriers of exposed collagen fibrils (organic matrix) induced by continuous and progressive mineral loss on the eroded surface hinder adequate adhesive infiltration [1,2,5]. Collagen fibrils that are not impregnated by adhesives become susceptible to hydrolytic degradation and create areas rich in water at the hybrid layer, promoting interfacial defects [2,5,6]. Additionally, bonding procedures can be impaired by an increase in accelerated erosive demineralization in the presence of pepsins, matrix metalloproteinases (MMPs), and cathepsins [7], from saliva and dentin [8], limiting the durability of the bonding interface.

With the continuous loss of tissue, the structural and aesthetic integrity of the teeth may be compromised, resulting in functional and aesthetic problems that require restorative intervention [9,10]. Thus, alternatives have been developed to increase the bond strength of eroded dentin (bur abrasion, sodium hypochlorite, and collagen cross-linker primers) [1,4,11,12]. However, there is still no consensus concerning which is the most reliable [1,4,11,12], especially as some of these protocols used experimental primers [12]. Although these protocols have been used [1,4,11,12], none prevented the degradation of eroded dentin over time.

Several clinical alternatives used for sound dentin [13] have not been tested for eroded dentin. It has been reported that short- and long-term resin–dentin bonding of universal adhesives can be improved by an additional hydrophobic resin coat [14,15,16,17,18]. The application of an additional hydrophobic resin coat aims to increase the thickness and uniformity of the adhesive layer and to reduce fluid flow across the adhesive interface [14,15,16,17,19,20]. This less permeable layer can help prevent the degradation of eroded dentin. Simplified adhesives, such as universal adhesives that combine hydrophilic and hydrophobic monomers in a unique bottle, promote the creation of an adhesive interface that lacks a non-solvated hydrophobic resin coating [21]. The formed hybrid layer is highly permeable to water from the oral environment, and to water fluxes from dentinal tubules [21]. A more hydrophilic adhesive has a higher water sorption rate, resulting in fast hydrolytic degradation of the hybrid layer [21,22,23,24].

Several researchers have advocated the use of an additional hydrophobic resin coat to improve the bonding performance of adhesives [18,25,26,27]. However, it must be considered that an eroded dentin surface presents a great challenge for dental adhesion [9,28] and that no previous study has evaluated the application of an additional hydrophobic resin coat on an eroded dentin surface.

Therefore, this in vitro study aims to evaluate the effect of an additional hydrophobic resin coat to improve the microtensile bond strength (less loss of restoration from a clinical point of view) and decrease the nanoleakage (less restorations with marginal discoloration from a clinical point of view) on eroded dentin bonding, when compared to sound dentin, after two years of water storage in comparison with immediate time.

The null hypotheses tested were as follows: (1) the use of an additional hydrophobic resin coat associated to universal adhesives would not affect microtensile bond strength (μTBS) or nanoleakage (NL) values when compared to only universal adhesive application; (2) these microtensile bond strength and nanoleakage values would not result in differences when adhesives would be evaluated on sound vs. eroded dentin and; (3) aging (immediate or after 2 years of water storage) would not affect microtensile bond strength or nanoleakage values.

## 2. Materials and Methods

### 2.1. Tooth Selection and Preparation

Sixty-four human molars were considered in this study. The teeth were collected after approval from the local ethics committee (#4.310.655). They were disinfected with 0.5% chloramine and stored in distilled water until use. The occlusal third of the crown was removed from all teeth using a diamond saw in a cutter machine with water-cooling (Isomet, Buehler, Lake Bluff, IL, USA) to obtain a flat dentin surface. To confirm the absence of enamel on the dentin surface, careful examination was performed under a stereomicroscope (Olympus SZ40, Tokyo, Japan) at 30× magnification. The exposed dentin surfaces were polished with wet #600-grit silicon carbide abrasive paper (SiC) for 30 s to standardize the smear layer.

### 2.2. Experimental Design

The teeth were randomly divided into 8 groups (*n* = 8 dentin specimens) using a combination of the following variables: 1. dentin substrate, including sound and eroded dentin; 2. treatment, including the control (adhesives applied according to manufacturer recommendations) and extra HL (control plus additional hydrophobic resin coat) and 3. storage time (24 h and after 2 years). Two universal adhesives (Prime & Bond Active (PBA, Dentsply Sirona, Charlotte, NC, USA) or Scotchbond Universal (SBU, 3M Oral Care, St Paul, MN, USA) were used and the specimens obtained for each tooth were randomly divided and tested (half after 24 h and half after 2 years of storage in water at 37 °C). Product information and application mode details for the experimental groups are provided in Table 1 and Table 2, respectively.

### 2.3. Sample Size Calculation

The sample size calculation was performed online (www.sealedenvelope.com, accessed on 24 February 2022. The sample size was determined using the microtensile bond strength (μTBS) mean ± standard deviation values for Scotchbond Universal on sound dentin reported in the literature (49.8 ± 5.3 MPa) [2,29,30]. To detect a difference of 8 MPa between the tested groups at a significance level of 5%, with a power of 80% and using a two-sided test, the minimum sample size was 8 teeth per group in accordance with the guidance on microtensile bond strength testing of dental composite bonding. [31]

### 2.4. pH Cycling Model

Thirty-two prepared human molars were randomly selected to simulate erosive demineralization. Before erosive cycling, the lateral and root areas were covered with two layers of nail varnish to allow erosive demineralization only on the occlusal surface. The specimens were exposed to an erosive cyclic demineralization and remineralization procedure by immersion in a soft drink (Coca-Cola, pH 2.6) 4 times daily for 90 s each (10 mL per specimen) for 5 days [2,7,32]. The soft drink was replaced for each immersion. After each demineralization, the specimens were rinsed with deionized water for 10 s and immersed in a remineralizing solution (4.08 mM H_3_PO_4_, 20.10 mM KCl, 11.90 mM Na_2_CO_3_, and 1.98 mM CaCl_2_, pH of 6.7, 10 mL per specimen) for 60 min [2,33]. The remineralization solution was replaced daily. The pH levels of all solutions were monitored periodically using a pH meter. Then, all teeth were thoroughly rinsed with water and the surrounding enamel was removed using a diamond bur in a high-speed handpiece (#2135, KG Sorensen; São Paulo, SP, Brazil) under water irrigation.

### 2.5. Restorative Procedures

The universal adhesives were only applied in the self-etch mode and according to the manufacturer instructions (Table 2). For all specimens, the dentin was kept visibly moist and the adhesive was applied and light-cured for 10 s at 1400 mW/cm^2^ (Valo, Ultradent Product, Salt Lake City, UT, USA) [14,15] in accordance with the manufacturer instructions (Table 2). For the extra HL groups, Clearfil SE Bond (Kuraray Noritake, Tokyo, Japan) was used (Table 2).

All teeth were restored using a composite resin buildup (Opallis, A2, FGM, Joinville, SC, Brazil) applied in 2 mm increments and each increment was light-cured for 40 s (1400 mW/cm^2^, Valo, Ultradent Product, Salt Lake City, UT, USA). A single trained operator performed all restorative procedures (Table 2).

The restored teeth were stored in distilled water at 37 °C for 24 h. Specimens were cut longitudinally using a cutting machine (Isomet, Buehler, Lake Bluff, IL, USA), and rotated in 90° angles to obtain resin–dentin bonded sticks with a cross-sectional area of approximately 0.8 mm², measured using digital calipers (Digimatic Caliper, Mitutoyo, Tokyo, Japan) to calculate the bond strength in MPa. All resin–dentin bonded sticks that underwent pretest debonding during specimen preparation were recorded for each tooth.

A total of 26–32 resin–dentin bonded sticks were obtained per tooth including the pretests debonding. The resin–dentin bonded sticks were divided as follows: for the nanoleakage test, 3 resin–dentin bonded sticks per tooth from each experimental condition group were tested after 24 h or 2 years of water storage; for microtensile bond strength, the remaining resin–dentin bonded sticks were tested after 24 h or 2 years of water storage. The distilled water was changed monthly.

### 2.6. Microtensile Bond Strength Test (μTBS)

After 24 h or 2 years of water storage, the resin–dentin bonded sticks were attached to a modified Geraldeli device [34] using a cyanoacrylate resin and subjected to tensile force in a universal testing machine (Katros Dinamometros, Cotia, SP, Brazil) at 0.5 mm/min, until bond failure occurred. Microtensile bond strength values were calculated by dividing the load at failure by the cross-sectional bonding area.

The failure mode of each resin–dentin bonded stick was observed using a digital microscope (Olympus SZ40, Tokyo, Japan) and classified as cohesive ((C), failure exclusively within the dentin or resin) or adhesive/mixed ((A/M), adhesive or mixed failure inside any of the bonded substrates). For statistical analysis, specimens with pre-test failures (PF) were included in the tooth mean with a value of 4.0 MPa [35].

### 2.7. Nanoleakage (NL)

Three resin–dentin bonded sticks per tooth for each storage condition that had not been used in the microtensile bond strength test were placed in an ammoniacal silver nitrate solution in the dark for 24 h, rinsed in distilled water, and immersed in a photo-developing solution for 8 h under fluorescent light [36,37]. The specimens were polished with 2500-grit SiC paper and 1-mm and 0.25-mm diamond paste (Buehler Ltd., Lake Bluff, IL, USA). After ultrasonic cleaning and air-drying, the specimens were mounted on stubs, coated with carbon-gold, and the silver penetration levels at the resin–dentin interface of each specimen were analyzed using a field-emission scanning electron microscope in backscattering mode (VEGA 3 TESCAN, Shimadzu, Tokyo, Japan).

Three images of each bonded stick were captured, including one at 0.3 mm to the right of center, one at 0.3 mm to the left of center, and one at the center. ImageJ software was used to determine the relative nanoleakage percentages along the adhesive and hybrid layers in each specimen [38].

### 2.8. Statistical Analysis

The microtensile bond strength and nanoleakage data for all the resin–dentin bonded sticks from the same hemi-tooth were averaged for statistical purposes. Thus, the experimental unit in this study was the hemi-tooth. After evaluating the normality (Kolmogorov–Smirnov) and equality of variances (Bartlett), the microtensile bond strength (MPa) and nanoleakage (%) data for each adhesive were subjected to three-way repeated measures ANOVA (dentin vs. treatment vs. storage time) and Tukey’s test. The level of significance was set at 5%. All analyses were performed using SPSS^®^ (Statistical Package for the Social Sciences), version 17.0 (SPSS Inc., Chicago, IL, USA).

## 3. Results

### 3.1. Microtensile Bond Strength (μTBS)

Approximately 80–104 resin–dentin bonded sticks per experimental group were evaluated for microtensile bond strength. The most common failure pattern in all the experimental groups was the adhesive/mixed-type failure (Figure 1). Few premature failures (0.6%) were observed after 24 h. After 2 years, 4.2% of the failures were considered premature failures. However, a significant increase was observed in the control group (Figure 1). The microtensile bond strengths values are presented in Table 3 A significant difference was only observed for the triple cross-product interaction (dentin vs. treatment vs. time) (Table 3; *p* < 0.000001), and for the main factors’ dentin, treatment, and time (*p* < 0.000001).

For Prime & Bond Active and Scotchbond Universal adhesives, extra HL did not significantly increase the microtensile bond strength values in the immediate group when applied to sound and eroded dentin, with the exception of Prime & Bond Active in eroded dentin (Table 3; *p* > 0.05).

For Prime & Bond Active, after two years of water storage, a significant decrease in the microtensile bond strength was observed for the control and extra HL groups, compared to the immediate results in both substrates (Table 3; *p* = 0.0001). In contrast, Scotchbond Universal did not significantly decrease the microtensile bond strength after two years of water storage, compared with the immediate results for the extra HL group (Table 3; *p* < 0.05).

After two years, the application of extra HL produced significantly higher microtensile bond strength values for both adhesives compared to the control group (Table 3; *p* = 0.0001). In all cases, the microtensile bond strength values for sound dentin were higher than those for eroded dentin (Table 3; *p* = 0.000001).

### 3.2. Nanoleakage (NL)

For nanoleakage, 24 resin–dentin bonded sticks per experimental group was evaluated. Nanoleakage data are presented in Table 4. A significant difference was observed in the triple cross-product interaction (dentin vs. treatment vs. time) (Table 4; *p* < 0.00001), and in the main factors’ dentin, treatment, and time (*p* < 0.0001).

For Prime & Bond Active and Scotchbond Universal adhesives, using extra HL did not significantly decrease silver nitrate uptake values in the immediate group when applied to sound dentin (Table 4; *p* > 0.05). In contrast, using extra HL on eroded dentin significantly decreased silver nitrate uptake values in the immediate group for both adhesives (Table 4; *p* = 0.0001).

After two years of water storage, in both substrates and for both adhesives, a significant increase in the silver nitrate uptake values was observed in the control groups, compared with the immediate results (Table 4; *p* = 0.0001). However, when an extra HL coat was applied, a significant increase in the silver nitrate uptake values compared with the immediate group was only observed for Prime & Bond Active on eroded dentin (Table 4; *p* = 0.0001).

After two years, the application of extra HL resulted in significantly lower silver nitrate uptake values than in the control group for both adhesives (Table 4; *p* = 0.0001). In all cases, the silver nitrate uptake values for sound dentin were lower than those for eroded dentin (Table 4; *p* = 0.000001).

## 4. Discussion

According to the results of the present study, the first null hypothesis was rejected, because the use of an additional hydrophobic resin coat increases microtensile bond strength and decreases nanoleakage values when compared to the use of universal adhesive only. The second null hypothesis was rejected, as the mean microtensile bond strength and nanoleakage values were lower in eroded dentin when compared to sound dentin. Finally, the third null hypothesis was also rejected, since after 2 years of water storage, lower microtensile bond strength and higher nanoleakage values were observed when compared to immediate time.

Despite the fact that all universal adhesives could be applied in the etch-and-rinse and self-etch mode, it is well-known that the self-etch strategy is preferred, mainly when eroded dentin is used as the substrate [39,40,41]. However, for both universal adhesives, higher bond strength and lower silver nitrate uptake values were observed for sound dentin than for eroded dentin. As mentioned in the introduction, erosive demineralization promotes the dissolution of the mineral component and the continued progression induces the formation of a dense, fibrous collagen network with buffering properties [42]. In addition, an increased loss of collagen periodicity occurs in the collagen matrix [43], and the spaces between the collagen fibrils are occupied by water [8].

These structural differences between sound and eroded dentin explain the additional problems in obtaining reliable bonding on eroded dentin; the eroded dentin structure influences the infiltration and polymerization of adhesive monomers [1,5,44] and lowers bond strength values, as reported in the literature [2,3,4,11].

Silver nitrate uptake on the bonding interface was more evident in eroded dentin, reflecting the presence of water-rich zones and indicating the inconsistent resin infiltration of the demineralized collagen (Figure 2). These features lead to the formation of a structurally imperfect and highly porous hybrid layer [45], resulting in areas of hydrophilic predominance and demineralized zones with collagen fibrils that are incompletely encapsulated by resin monomers [6,46], contributing to the reduced bonding performance of universal adhesives on eroded dentin, as reported in previous studies [2,3,4,11].

In the immediate period, the use of extra HL resulted in a significant decrease in the silver nitrate uptake values in eroded dentin compared to sound dentin. It is known that eroded dentin has a greater water content [8], which can hinder adequate adhesive infiltration [2,5], but also leads to more liquid retention in the highly hydrophilic and porous adhesive layer, as achieved with simplified adhesives [47]. The application of extra HL on eroded dentin seems to limit the diffusion of water through the hybrid layer to the adhesive interface [19,36], in addition to increasing the degree of polymerization of a simplified adhesive and decreasing its immediate permeability [13,48].

After two years, the application of extra HL resulted in significantly higher bond strength and lower silver nitrate uptake values than the control group for both adhesives in sound and eroded dentin. Universal adhesives are considered as one-step simplified adhesives, due to the presence of hydrophobic and hydrophilic components mixed with organic solvents, without a separate hydrophobic as a final coat [14,49]. It is known that complete solvent elimination does not occur for a highly hydrophilic adhesive [50,51], and the presence of residual volatile solvents may prevent approximation between reactive pendant species [52,53], directly influencing the conversion degree of hybrid and adhesive layers [13,48]. As a result, the hybrid layer formed for simplified adhesives can behave as a permeable membrane [36,54,55] that allows for bidirectional water movement across the adhesive layer [29].

Thus, extra HL applied over such adhesive systems provides additional free radicals to enhance the rate and extent of polymerization of simplified adhesives, with an expected increase in the bond strength to dentin [48,52,53]. Furthermore, thickening of the adhesive layer has been shown to improve dentin bonding once the interface permeability is reduced [52,56,57]. Thus, the use of extra HL makes these adhesives less prone to hydrolytic degradation processes, as the resultant adhesive interface is more hydrophobic, with decreased water sorption through osmosis from the underlying dentin in the long term [58,59,60], especially in eroded dentin, which contains a significant amount of water compared to sound dentin.

Previous studies on sound dentin indicated that bonding performance improves with increased thickness of the adhesive layer; increased bond strengths were achieved by applying multiple adhesive coats. This is relevant as universal adhesives commonly have a thin film thickness (<10 μm) [61]. Once the use of extra HL indicated beneficial results for sound dentin, it was expected that universal adhesives light-cured before application of the extra layer on eroded dentin may have thickened the adhesive layer and improved aging resistance [14,16,17,26], as confirmed in this study.

Although there was a significant decrease in bond strength after two years in the control and extra HL groups for Prime & Bond Active, compared to the immediate results, a significant decrease was not observed for Scotchbond Universal with extra HL. One difference between Scotchbond Universal and Prime & Bond Active is the presence of a polyalkenoic acid copolymer in Scotchbond Universal. Initially, the rationale for using the polyalkenoic acid copolymer was to provide better moisture stability [62]. However, more recently, it was observed that the carboxyl groups present in polyalkenoic acids replace the phosphate ions in hydroxyapatite, establishing ionic bonding with calcium [63], preventing or decreasing degradation in in vitro conditions, as observed by Sezinando et al. [64].

It was observed that, despite the additional hydrophobic resin layer that indicated reduced silver nitrate uptake values and improvements in bond strength, which allowed the formation of a more durable resin–eroded dentin interface, some degradation of the hybrid layer was still observed after two years of water storage. Therefore, another simple means of improving the adhesive properties of hydrophobic coatings is to incorporate bioactive materials in their contents, including different bioglass, or even more promising materials, such as phosphorene and borophene [65,66,67,68]. However, future research should evaluate the effects of a hydrophobic resin layer containing bioactive materials on the resin–eroded adhesive interface in the long term.

It is important to mention some limitations of the present study. One of them is related to the fact that, despite the promising results observed in the present study, this is an in vitro study, which only partially simulated the intraoral conditions. Therefore, future clinical studies evaluating the effect of an additional hydrophobic resin layer associated with universal adhesives in the restoration of eroded teeth should be carried out. The second limitation is the fact that only two universal adhesives were evaluated. As universal adhesives could be considered a class of materials with several differences regarding their composition [69], future studies with other universal adhesives need to be conducted to prove if the use of an additional hydrophobic resin layer could produce the same results observed in the present study.

## 5. Conclusions

An additional hydrophobic resin coating increased the bond strength and reduced nanoleakage in eroded and sound dentin in the immediate time when compared to the application of universal adhesives only. However, this additional hydrophobic resin coat significant decreased the degradation of dentin after 2 years of water storage, mainly for eroded dentin, the most degradable substrate. Therefore, this strategy could be considered a feasible alternative to improve the adhesive properties of eroded dentin after two years of water storage. Further clinical studies or hydrophobic coatings containing bioactive materials are needed as alternatives to improve the adhesive properties of eroded dentin.

## Figures and Tables

**Figure 1 polymers-14-02701-f001:**
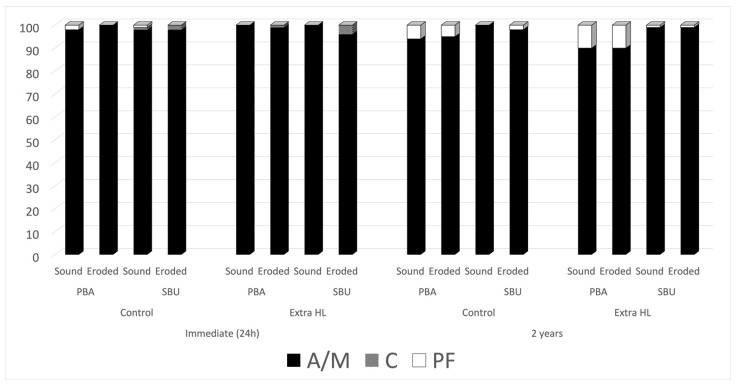
Fracture mode of specimens for all experimental conditions. Abbreviations: A/M, adhesive/mixed fracture mode, C, cohesive fracture mode, PF, premature failures.

**Figure 2 polymers-14-02701-f002:**
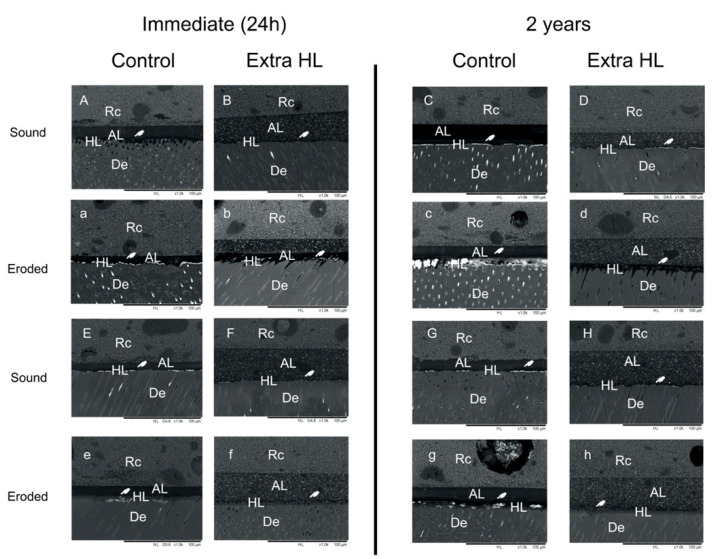
Representative backscatter scanning electron microscope micrographs of adhesive interface for all experimental groups (1.0 kx). Silver nitrate deposits were detected in all groups, mainly in the hybrid layer (white hands). Generally, sound dentin (capital letters) exhibited less silver nitrate infiltration than eroded dentin (lowercase letters) at both evaluation times. Overall, reduced silver nitrate uptake was observed in sound dentin and eroded dentin with extra HL application for both adhesives in the immediate evaluation. However, silver nitrate infiltration increased significantly in the control group compared to the extra HL group after two years of water storage.

**Table 1 polymers-14-02701-t001:** Material, batch number and composition of the materials used.

Material	Batch Number	Composition
Clearfil SE Bond Kuraray Noritake (Extra HL)	5U0640	Only Bond bottle: 10-MDP, Bis-GMA, hydrophobic dimethacrylate, HEMA, CQ, *N*,*N*-diethanol p-toluidine, colloidal silica
Prime & Bond Active Dentsply Sirona (PBA)	2009000399	Bisacrylamide 1 (25–50%), 10-MDP (10–25%), bisacrylamide 2 (2.5–10%), 4-(dimethylamino) benzonitrile (0.1–1%), PENTA, propan-2-ol (10–25%), water (20%).
Scotchbond Universal 3M Oral Care (SBU)	2019100137	10- MDP, dimethacrylate resins, Bis-GMA, HEMA, methacrylatemodified polyalkenoic acid copolymer, CQ, filler, ethanol, water, initiators, silane.

Abbreviations: Bis-GMA: bisphenol A diglycidylmethacrylate; CQ: canforquinone; HEMA: 2-hydroxyethyl methacrylate; PENTA: dipentaerythritol penta-acrylate phosphate; 10-MDP: methacryloyloxydecyldihydrogen phosphate.

**Table 2 polymers-14-02701-t002:** Application mode of different universal adhesives for both dentinal substrates.

Adhesive System	Experimental Groups	Application Mode *
Prime & Bond Active	Control	Apply the adhesive to the entire preparation with a microbrush and rub it in for 20 s. Apply a gentle stream of air over the liquid for at least 5 s. Light-cure for 10 s at 1400 mW/cm^2^.
	Extra HL	Apply the adhesive to the entire preparation with a microbrush and rub it in for 20 s. Apply a gentle stream of air over the liquid for at least 5 s. Light cure for 10 s at 1400 mW/cm^2^. Apply a very thin layer of extra HL with a microbrush Air blow to achieve an optically thin layer. Light cure for 10 s at 1400 mW/cm^2^.
Scotchbond Universal	Control	Apply the adhesive to the entire preparation and leave undisturbed for 20 s. Direct a gentle stream of air over the liquid for about 5 s until it no longer moves, and the solvent evaporates completely. Light-cure for 10 s at 1400 mW/cm^2^.
	Extra HL	Apply the adhesive to the entire preparation and leave undisturbed for 20 s. Direct a gentle stream of air over the liquid for about 5 s until it no longer moves, and the solvent evaporates completely. Light cure for 10 s at 1400 mW/cm^2^. Apply a very thin layer of extra HL with a microbrush. Air blow to achieve an optically thin layer. Light cure for 10 s at 1400 mW/cm^2^.

* The materials were applied according to the recommendations of the respective manufacturers only in the self-etch mode.

**Table 3 polymers-14-02701-t003:** Mean (in MPa) ± standard deviations of microtensile bond strength for all experimental conditions, as well as statistical analyses.

Experimental Groups		Immediate (24 h)		2 Years	
		Control	Extra HL	Control	Extra HL
PBA	Sound	42.9 (4.5) A,B	48.2 (4.2) A	19.5 (3.9) D,E	39.1 (4.1) B
	Eroded	32.1 (4.2) C	38.0 (4.8) B	15.2 (3.0) E	35.1 (3.9) B,C
SBU	Sound	46.5 (4.1) a,b	51.2 (3.9) a	21.9 (2.3) e	45.8 (4.0) a,b
	Eroded	28.1 (3.9) c,d	39.7 (3.5) b,c	15.6 (3.3) e	33.2 (3.9) c

Different capital or lower case letters mean statistically significant difference among groups for each adhesive (3-way ANOVA and Tukey’s test; *p* = 0.05).

**Table 4 polymers-14-02701-t004:** Mean (in %) ± standard deviations of nanoleakage for all experimental conditions, as well as statistical analyses.

Experimental Groups		Immediate (24 h)		2 Years	
		Control	Extra HL	Control	Extra HL
PBA	Sound	8.7 (1.5) A,B	6.4 (1.8) A	16.9 (2.0) B,C	9.7 (1.5) B
	Eroded	19.8 (2.5) C	14.0 (2.4) B	26.3 (3.0) D	18.1 (2.3) C
SBU	Sound	6.6 (1.7) a	7.8 (1.5) a	14.5 (1.4) b	8.4 (1.7) a
	Eroded	19.2 (1.6) c	13.6 (1.7) b	30.0 (1.4) d	16.4 (2.7) b,c

Different capital or lower case letters mean statistically significant difference among groups for each adhesive (3-way ANOVA and Tukey’s test; *p* = 0.05).

## Data Availability

The data presented in this study are available on request from the corresponding author.

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
