# Peer review of "Does the Application of Additional Hydrophobic Resin to Universal Adhesives Increase Bonding Longevity of Eroded Dentin?"

_polymers, 2022, doi:10.3390/polym14132701_

Round 1

Reviewer 1 Report

Despite this paper may have some interest, in the current form it does not meet the minimum criteria for publication.

It should improve introduction with other similar studies, with other different but related studies and with one or more translational aims to be applied into the daily clinical practice.

Methods have serious concerns, as the overall number of samples may be enough, but the experimental groups are too many: this strongly reduce the number of sample-per-group.

Results should be represented in a more informative and clear way (figures should be added together with graphics).

Conclusions must be improved.

Limits should be highlighted.

Author Response

Review #1

Despite this paper may have some interest, in the current form it does not meet the minimum criteria for publication.

  1. It should improve introduction with other similar studies, with other different but related studies and with one or more translational aims to be applied into the daily clinical practice.

AU: We added several references in the introduction section, as well as two translational aims, in accordance with the reviewer suggestions.

  1. Methods have serious concerns, as the overall number of samples may be enough, but the experimental groups are too many: this strongly reduce the number of sample-per-group.

AU: We understand the concern of reviewer about the number samples. However, taking in consideration a recent guidance published regarding the microtensile bond strength test, that recommend between 8 to 10 teeth for experimental group [1], it is possible to affirm that the number of samples are enough, because 8 teeth were used by experimental group. Also, this number of teeth was based on a sample size calculation for the primary outcome of the present study.

Actually, as several micro specimens are obtained by each tooth, it was possible to calculate that, in the present study, among 26 – 32 resin-dentin bonded sticks per tooth were obtained. This means that approximately 80-104 resin-dentin bonded sticks per experimental group was evaluated at the end in the microtensile bond strength. Regarding to nanoleakage (secondary outcome), the same number of teeth was used (n = 8) and 3 resin-dentin bonded sticks per tooth and per experimental condition were obtained. This means that 24 resin-dentin bonded sticks per experimental group was evaluated at the end. All these explanations were adding in the present version of the manuscript to clarify the sample size for the readers of Polymers.

  1. Results should be represented in a more informative and clear way (figures should be added together with graphics).

AU: The fracture pattern data was modified in accordance with the reviewer suggestion.

  1. Conclusions must be improved.

AU: This section was improved in the present version of manuscript.

  1. Limits should be highlighted.

AU: Some limitations were adding in the present version of the manuscript.

REFERENCE:

1.         Armstrong, S.; Breschi, L.; Ozcan, M.; Pfefferkorn, F.; Ferrari, M.; Van Meerbeek, B. Academy of Dental Materials guidance on in vitro testing of dental composite bonding effectiveness to dentin/enamel using micro-tensile bond strength (muTBS) approach. Dent Mater 2017, 33, 133-143, doi:10.1016/j.dental.2016.11.015.

Reviewer 2 Report

Since it is a ready-made material and measurement method, it lacks novelty.

There are few bonding agents.

There is no novelty in the consideration.

Although it is acceptable for a long period of 2 years, it is not particularly novel because it is a natural result if it is used in combination with phosphoric acid treatment.

Author Response

Review #2

  1. Since it is a ready-made material and measurement method, it lacks novelty.

AU: We understand the concerns of the reviewer in relation to the innovation of material and methodology used. However, it's important to mention the following factors: 1) considering that eroded tooth it’s a common phenomenon in the general population in developed countries [2], with a prevalence between 20% - 45% [3,4]; 2) considering that restorative treatment it's an alternative to reduce or stop the erosive progression lesion, at the same time that reduce symptoms of pain or dentine hypersensitivity [4]; and 3) considering that dental erosion involves complex histological dentinal changes that affect the quality and durability of the eroded-dentin interface [5-8], clinical alternatives frequently used in sound dentin [9-15]should be tested in eroded dentin in order to achieve and maintain successful adhesive restorative treatment. Therefore, we believe that these argues clarify the originality and relevance of the present study.

  1. There are few bonding agents.

AU: The reviewer is right. Even though we only tested two universal adhesives frequently used in dentistry with different chemical compositions, we agree with the reviewer that the number of adhesives used it’s a limitation and this has been reported in the present version of the manuscript.

  1. There is no novelty in the consideration.

AU: According to described in the response to the first question, we believe that all argues make clear the originality and relevance of the present study and we urge to reviewer can accept our argues.

  1. Although it is acceptable for a long period of 2 years, it is not particularly novel because it is a natural result if it is used in combination with phosphoric acid treatment.

AU: It’s not clear for the authors, which exactly the reviewer refers to. Actually, to extent of author's knowledge, it was not found any study previously published evaluating the effect of an additional hydrophobic resin coat associated to universal adhesive and evaluated after long-term on eroded dentin, according to previously described. Observe that, in the first version of the present study, the results were presented evaluating universal adhesives in both etching mode (with and without combination with phosphoric acid) and no significant difference were observed when different adhesive strategies were evaluated. However, the reviewer 3 suggested that only self-etch results should be presented and in accordance with this comment, in the present version of the manuscript only self-etch results were added.

REFERENTES:

  1. Not applied (cited in response to Reviewer 1)
  2. Schlueter, N.; Amaechi, B.T.; Bartlett, D.; Buzalaf, M.A.R.; Carvalho, T.S.; Ganss, C.; Hara, A.T.; Huysmans, M.; Lussi, A.; Moazzez, R.; et al. Terminology of Erosive Tooth Wear: Consensus Report of a Workshop Organized by the ORCA and the Cariology Research Group of the IADR. Caries Res 2020, 54, 2-6, doi:10.1159/000503308.
  3. Schlueter, N.; Luka, B. Erosive tooth wear - a review on global prevalence and on its prevalence in risk groups. Br Dent J 2018, 224, 364-370, doi:10.1038/sj.bdj.2018.167.
  4. Carvalho, T.S.; Colon, P.; Ganss, C.; Huysmans, M.C.; Lussi, A.; Schlueter, N.; Schmalz, G.; Shellis, P.R.; Bjorg Tveit, A.; Wiegand, A. Consensus Report of the European Federation of Conservative Dentistry: Erosive tooth wear diagnosis and management. Swiss Dent J 2016, 126, 342-346.
  5. de Rossi, G.R.C.; Ozcan, M.; Volpato, C.A.M. How to improve bond stability to eroded dentin: a comprehensive review. Journal of Adhesion Science and Technology 2021, 35, 1015-1034, doi:10.1080/01694243.2020.1835266.
  6. Siqueira, F.S.F.; Cardenas, A.M.; Ocampo, J.B.; Hass, V.; Bandeca, M.C.; Gomes, J.C.; Reis, A.; Loguercio, A.D. Bonding performance of universal adhesives to eroded dentin. J Adhes Dent 2018, 20, 121-132, doi:10.3290/j.jad.a40300.
  7. Siqueira, F.; Cardenas, A.; Gomes, G.M.; Chibinski, A.C.; Gomes, O.; Bandeca, M.C.; Loguercio, A.D.; Gomes, J.C. Three-year effects of deproteinization on the in vitro durability of resin/dentin-eroded interfaces. Oper Dent 2018, 43, 60-70, doi:10.2341/16-308-L.
  8. Zimmerli, B.; De Munck, J.; Lussi, A.; Lambrechts, P.; Van Meerbeek, B. Long-term bonding to eroded dentin requires superficial bur preparation. Clin Oral Investig 2012, 16, 1451-1461, doi:10.1007/s00784-011-0650-8.
  9. Loguercio, A.D.; Reis, A. Application of a dental adhesive using the self-etch and etch-and-rinse approaches: an 18-month clinical evaluation. J Am Dent Assoc 2008, 139, 53-61, doi:10.14219/jada.archive.2008.0021.
  10. Munoz, M.A.; Sezinando, A.; Luque-Martinez, I.; Szesz, A.L.; Reis, A.; Loguercio, A.D.; Bombarda, N.H.; Perdigao, J. Influence of a hydrophobic resin coating on the bonding efficacy of three universal adhesives. J Dent 2014, 42, 595-602, doi:10.1016/j.jdent.2014.01.013.
  11. Perdigao, J.; Ceballos, L.; Giraldez, I.; Baracco, B.; Fuentes, M.V. Effect of a hydrophobic bonding resin on the 36-month performance of a universal adhesive-a randomized clinical trial. Clin Oral Investig 2020, 24, 765-776, doi:10.1007/s00784-019-02940-x.
  12. Reis, A.; Carrilho, M.; Breschi, L.; Loguercio, A.D. Overview of clinical alternatives to minimize the degradation of the resin-dentin bonds. Oper Dent 2013, 38, E1-E25, doi:10.2341/12-258-LIT.
  13. Sezinando, A.; Luque-Martinez, I.; Munoz, M.A.; Reis, A.; Loguercio, A.D.; Perdigao, J. Influence of a hydrophobic resin coating on the immediate and 6-month dentin bonding of three universal adhesives. Dent Mater 2015, 31, e236-246, doi:10.1016/j.dental.2015.07.002.
  14. Ahmed, M.H.; De Munck, J.; Van Landuyt, K.; Peumans, M.; Yoshihara, K.; Van Meerbeek, B. Do universal adhesives benefit from an extra bonding layer? J Adhes Dent 2019, 21, 117-132, doi:10.3290/j.jad.a42304.
  15. Ahmed, M.H.; Yao, C.; Van Landuyt, K.; Peumans, M.; Van Meerbeek, B. Extra Bonding Layer Compensates Universal Adhesive's Thin Film Thickness. J Adhes Dent 2020, 22, 483-501, doi:10.3290/j.jad.a45179.

Reviewer 3 Report

Comments to the authors:

Please find bellow some suggestions that could improve your work.

Title

  1. Title grammar must be improved.

Abstract section

  1. Please include results related to the factors Dentin substrate, Universal adhesives, and Adhesive strategies. They were included in the statistical analysis but results are not presented here.

Introduction section

  1. What do you mean with the term “continuous acid”?

Materials and methods section

  1. I would suggest to simplify your experimental design by deleting the factor “adhesive strategy” from your analysis. It’s well known that the self-etch strategy is preferred when dentin is used as substrate.
  2. Actually, your hypothesis only includes the use of an additional hydrophobic resin factor. So, why you decide to include the other factors?
  3. With regards to the statistical analysis, a four-way ANOVA test was evaluated. Considering this, the factor adhesive was not included in the analysis (i.e. each adhesive was analyzed separately). However, this is not stated in the abstract section. Please clarify.

Results section

  1. I suggest to change the symbol ± by including the standard deviation between parenthesis. This change will easy the reading and understanding of the tables.

Discussion section

  1. Please expand your discussion section. Not all the factors studied are included.

Conclusion section

  1. Consider the other factors studied too, not only the factor related to the hydrophobic layer.

Author Response

Review #3

Comments to the authors:

Please find bellow some suggestions that could improve your work.

Title

  1. Title grammar must be improved.

A: Thank you for the suggestion. We revised the title in the present version of manuscript.

Abstract section

  1. Please include results related to the factors Dentin substrate, Universal adhesives, and Adhesive strategies. They were included in the statistical analysis but results are not presented here.

AU: We added the results requested by the reviewer in the present version of manuscript.

Introduction section

  1. What do you mean with the term “continuous acid”?

AU: This term refers to “continuous acid action” that promoted erosive tooth wear. The word “action” was lost in the previous version. However, it was included in the present version of manuscript

Materials and methods section

  1. I would suggest to simplify your experimental design by deleting the factor “adhesive strategy” from your analysis. It’s well known that the self-etch strategy is preferred when dentin is used as substrate.

AU: The reviewer is right, and we deleted the factor “adhesive strategy” from our analysis.

  1. Actually, your hypothesis only includes the use of an additional hydrophobic resin factor. So, why you decide to include the other factors?

AU: The reviewer is right. We rewrote our hypothesis to properly described the different factors to be evaluated as following: “The null hypotheses tested were that: (1) use of an additional hydrophobic resin coat associated to universal adhesives would not affect microtensile bond strength (μTBS) or nanoleakage (NL) values when compared to only universal adhesive application; (2) these μTBS and NL values would not result in difference when adhesives would be evaluated in sound vs. eroded dentin and; (3) these μTBS and NL values would be evaluated in different period of time (immediate or after 2 years of water storage).”

  1. With regards to the statistical analysis, a four-way ANOVA test was evaluated. Considering this, the factor adhesive was not included in the analysis (i.e. each adhesive was analyzed separately). However, this is not stated in the abstract section. Please clarify.

AU: After accepting your suggestion in the question 4, it was run a three-way ANOVA (dentin vs. treatment vs. storage time). This new statistical analysis was inserted in the abstract section in the second version of the manuscript.

Results section

  1. I suggest to change the symbol ± by including the standard deviation between parenthesis. This change will easy the reading and understanding of the tables.

AU: Thank you for your suggestion. We modified according to reviewer suggestion.

Discussion section

  1. Please expand your discussion section. Not all the factors studied are included.

AU: After accepting your suggestion in the question 4, the discussion section was improved discussed all significant factors.

Conclusion section

  1. Consider the other factors studied too, not only the factor related to the hydrophobic layer.

AU: We agree with the reviewer and this section was modified in the present version of manuscript in according with the suggestion.

Round 2

Reviewer 1 Report

Authors have improved their paper.

However, major points need to be addressed.

i) Authors must also compare their approach with other promising biomaterials, such as the biomedical applications of several new biomaterials like the Phosphorene or the Borophene, independly to the surrent use in the reported matter. (Phosphorene Is the New Graphene in Biomedical Applications. Materials (Basel, Switzerland), 12(14), 2301) – AND - (Borophene Is a Promising 2D Allotropic Material for Biomedical Devices. Appl. Sci. 2019, 9, 3446. )

ii) Conclusions need to be increased with more related text and future insights on this matter.

iii) Limitations should be reported.

iv) Please explain all the acronyms throughout the text

Author Response

Thank you for the revision of the present manuscript. We have modified the manuscript, accordingly, taking into consideration all the points mentioned. All places in the manuscript where changes were made, we used the “track change" mode to indicate the corrections. A rebuttal letter responding point-by- point the concerns of each reviewer was prepared in order to facilitate the new revision process. Please see below:

  1. Authors must also compare their approach with other promising biomaterials, such as the biomedical applications of several new biomaterials like the Phosphorene or the Borophene, independly to the surrent use in the reported matter. (Phosphorene Is the New Graphene in Biomedical Applications. Materials (Basel, Switzerland), 12(14), 2301) – AND - (Borophene Is a Promising 2D Allotropic Material for Biomedical Devices. Appl. Sci. 2019, 9, 3446.)

A: Both references were added in the present version of the manuscript, according with the reviewer suggestion.

  1. Conclusions need to be increased with more related text and future insights on this matter.

A: The section was modified according with the reviewer suggestion.

  1. Limitations should be reported.

A: Limitations was related in the discussion section according with the reviewer suggestion.

  1. Please explain all the acronyms throughout the text

A: We modified the present version of manuscript according with the reviewer suggestion.

Reviewer 2 Report

This content has already been reported in various places and lacks novelty. Therefore, there is no originality.

Author Response

The authors understand the concern of reviewer. However, the authors of the present study understand “Novelty” as an unprecedented concept has been conceived (or unprecedented artifact has been produced). While “Originality” means an “originator” has synthesized a concept (or produced an artifact) by his own effort. This does not mean the concept/artifact is unprecedented in the world, only that it is unprecedented in the experience of the conceiver.

AU: The authors understand the concern of reviewer. However, the authors of the present study understand “Novelty” as an unprecedented concept/artifact has been conceived, that it was not performed in the present study. On the other side, “Originality” means an “originator” has created a concept/artifact by his own effort. This does not mean the concept/artifact is unprecedented in the world, only that it is unprecedented in terms of the new application. As previously described, in this sense, the authors believe that the present study has certain originality, mainly because this is the first one to evaluate this approach (hydrophobic coat) in eroded dentin. As described in the first reply, eroded tooth it’s a common phenomenon in the general population in developed countries [1], with a prevalence between 20% - 45% [2,3]; 2) considering that restorative treatment it's an alternative to reduce or stop the erosive progression lesion, at the same time that reduce symptoms of pain or dentine hypersensitivity [4]; and 3) considering that dental erosion involves complex histological dentinal changes that affect the quality and durability of the eroded-dentin interface [4-7], clinical alternatives frequently used in sound dentin [8-14] should be tested in eroded dentin in order to achieve and maintain successful adhesive restorative treatment. Therefore, we believe that these argues clarify the importance of publishing the present study.

References

  1. Schlueter, N.; Amaechi, B.T.; Bartlett, D.; Buzalaf, M.A.R.; Carvalho, T.S.; Ganss, C.; Hara, A.T.; Huysmans, M.; Lussi, A.; Moazzez, R.; et al. Terminology of Erosive Tooth Wear: Consensus Report of a Workshop Organized by the ORCA and the Cariology Research Group of the IADR. Caries Res 2020, 54, 2-6, doi:10.1159/000503308.
  2. Schlueter, N.; Luka, B. Erosive tooth wear - a review on global prevalence and on its prevalence in risk groups. Br Dent J 2018, 224, 364-370, doi:10.1038/sj.bdj.2018.167.
  3. Carvalho, T.S.; Colon, P.; Ganss, C.; Huysmans, M.C.; Lussi, A.; Schlueter, N.; Schmalz, G.; Shellis, P.R.; Bjorg Tveit, A.; Wiegand, A. Consensus Report of the European Federation of Conservative Dentistry: Erosive tooth wear diagnosis and management. Swiss Dent J 2016, 126, 342-346.
  4. de Rossi, G.R.C.; Ozcan, M.; Volpato, C.A.M. How to improve bond stability to eroded dentin: a comprehensive review. Journal of Adhesion Science and Technology 2021, 35, 1015-1034, doi:10.1080/01694243.2020.1835266.
  5. Siqueira, F.S.F.; Cardenas, A.M.; Ocampo, J.B.; Hass, V.; Bandeca, M.C.; Gomes, J.C.; Reis, A.; Loguercio, A.D. Bonding performance of universal adhesives to eroded dentin. J Adhes Dent 2018, 20, 121-132, doi:10.3290/j.jad.a40300.
  6. Siqueira, F.; Cardenas, A.; Gomes, G.M.; Chibinski, A.C.; Gomes, O.; Bandeca, M.C.; Loguercio, A.D.; Gomes, J.C. Three-year effects of deproteinization on the in vitro durability of resin/dentin-eroded interfaces. Oper Dent 2018, 43, 60-70, doi:10.2341/16-308-L.
  7. Zimmerli, B.; De Munck, J.; Lussi, A.; Lambrechts, P.; Van Meerbeek, B. Long-term bonding to eroded dentin requires superficial bur preparation. Clin Oral Investig 2012, 16, 1451-1461, doi:10.1007/s00784-011-0650-8.
  8. Loguercio, A.D.; Reis, A. Application of a dental adhesive using the self-etch and etch-and-rinse approaches: an 18-month clinical evaluation. J Am Dent Assoc 2008, 139, 53-61, doi:10.14219/jada.archive.2008.0021.
  9. Munoz, M.A.; Sezinando, A.; Luque-Martinez, I.; Szesz, A.L.; Reis, A.; Loguercio, A.D.; Bombarda, N.H.; Perdigao, J. Influence of a hydrophobic resin coating on the bonding efficacy of three universal adhesives. J Dent 2014, 42, 595-602, doi:10.1016/j.jdent.2014.01.013.
  10. Perdigao, J.; Ceballos, L.; Giraldez, I.; Baracco, B.; Fuentes, M.V. Effect of a hydrophobic bonding resin on the 36-month performance of a universal adhesive-a randomized clinical trial. Clin Oral Investig 2020, 24, 765-776, doi:10.1007/s00784-019-02940-x.
  11. Reis, A.; Carrilho, M.; Breschi, L.; Loguercio, A.D. Overview of clinical alternatives to minimize the degradation of the resin-dentin bonds. Oper Dent 2013, 38, E1-E25, doi:10.2341/12-258-LIT.
  12. Sezinando, A.; Luque-Martinez, I.; Munoz, M.A.; Reis, A.; Loguercio, A.D.; Perdigao, J. Influence of a hydrophobic resin coating on the immediate and 6-month dentin bonding of three universal adhesives. Dent Mater 2015, 31, e236-246, doi:10.1016/j.dental.2015.07.002.
  13. Ahmed, M.H.; De Munck, J.; Van Landuyt, K.; Peumans, M.; Yoshihara, K.; Van Meerbeek, B. Do universal adhesives benefit from an extra bonding layer? J Adhes Dent 2019, 21, 117-132, doi:10.3290/j.jad.a42304.
  14. Ahmed, M.H.; Yao, C.; Van Landuyt, K.; Peumans, M.; Van Meerbeek, B. Extra Bonding Layer Compensates Universal Adhesive's Thin Film Thickness. J Adhes Dent 2020, 22, 483-501, doi:10.3290/j.jad.a45179.

Reviewer 3 Report

  • Please correct the hypothesis number 3, I think that this hypothesis should be like this:

(3) aging (immediate or after 2 years of water storage) would not affect microtensile bond strength (μTBS) or nanoleakage (NL).

  • I suggest to change reference 34 by this one: https://doi.org/10.3290/j.jad.a41975

Author Response

Thank you for the revision of the present manuscript. We have modified the manuscript, accordingly, taking into consideration all the points mentioned. All places in the manuscript where changes were made, we used the “track change" mode to indicate the corrections. A rebuttal letter responding point-by- point the concerns of each reviewer was prepared in order to facilitate the new revision process. Please see below:

  1. Please correct the hypothesis number 3, I think that this hypothesis should be like this: (3)aging (immediate or after 2 years of water storage) would not affect microtensile bond strength (μTBS) or nanoleakage (NL).

A: Thanks for the suggestion. We modified the present version of the manuscript according with the reviewer suggestion.

  1. I suggest to change reference 34 by this one: https://doi.org/10.3290/j.jad.a41975

A: We modified the reference according with the reviewer suggestion.